# Conditional deletion of *Tmem63b* does not impact mouse voiding behavior

**Marianela G. Dalghi**[1], **Wily G. Ruiz**[1], **Dennis R. Clayton**[1], **Tanmay Parakala-Jain**[1], **Marcelo D. Carattino**[1,2], **Yun Stone Shi**[3], **Gerard Apodaca**[1,2]*

1 Department of Medicine Renal-Electrolyte Division, University of Pittsburgh School of Medicine, Pittsburgh, Pennsylvania, United States of America, 2 Department of Cell Biology, University of Pittsburgh School of Medicine, Pittsburgh, Pennsylvania, United States of America, 3 Ministry of Education Key Laboratory of Model Animal for Disease Study, Model Animal Research Center, Medical School, Nanjing University, Nanjing, China

* gla6@pitt.edu

## Abstract

The proper function of the lower urinary tract depends on its ability to sense and react to mechanical forces as urine is produced, transported, stored, and eliminated; however, our current understanding of the mechanosensors involved in these events is limited. TMEM63 ion channels are reported to function as mechanosensors/osmosensors in other organs, and our studies revealed that the primary site of *Tmem63a* and *Tmem63b* gene expression and TMEM63B protein expression in the mouse bladder wall was the urothelium. Despite this localization, voiding behavior in conditional urothelial *Tmem63b* knockout mice, assessed using a video-monitored void-spot screening assay, was not significantly different from control mice, even when the urothelium was stressed by exposure to cyclophosphamide. We further observed that dorsal root ganglia sensory neurons, including those innervating the bladder, were also sites of *Tmem63a*, *Tmem63b*, and TMEM63B expression. Again, voiding behavior was not impacted in conditional sensory neuron *Tmem63b* knockout mice, treated or not with cyclophosphamide. Our studies reveal that the urothelium and dorsal root ganglia are sites of *Tmem63a*, *Tmem63b*, and TMEM63B expression, but deletion of *Tmem63b* alone in these tissues does not result in a demonstrable voiding phenotype.

## Introduction

The normal function of the lower urinary tract depends on mechanisms to sense and respond to mechanical forces as urine traverses the renal pelvis, ureters, bladder, and urethra. These forces include: shear stress as newly formed urine is propelled across the mucosal surfaces of the ureter; tensile forces that arise in the lining urothelium and subjacent tissues as the bladder fills with urine; and compressive forces that are generated as the detrusor contracts and the mucosal surface of the

**Data availability statement:** The data, along with the original image file for the RT-PCR data presented in Figure 1A, is included in the following data repository (Figshare): 10.6084/m9.figshare.30357820. This DOI will become active upon publication.

**Funding:** This work was supported by grants from the National institutes of Health including R01DK119183 (to GA and MDC), R01DK129473 (to GA), a pilot project grant supported by P30DK079307 (to MGD), and by the Pittsburgh Center for Kidney Research KIDNIT imaging core (U54DK137329). The work was also supported by grants from the National Natural Science Foundation of China 32330044 (to YSS). The Leica Stellaris confocal used in this study was funded in large part by S10OD028596 (to GA). The funders had no role in study design, data collection and analysis, decision to publish, or preparation of the manuscript.

**Competing interests:** The authors have declared that no competing interests exist.

bladder refolds in response to voiding. Perturbations in these events may underlie or contribute to several lower urinary tract pathologies including neurogenic bladder, partial outlet obstruction, and conditions such as overactive bladder and underactive bladder. Although all tissues in the bladder wall are reported to be mechanosensitive [1–9], much research has focused on the interface between the urothelium and subjacent nerve processes. A current hypothesis is that bladder filling triggers the serosal release of urothelial mediators (e.g., ATP, adenosine, NO, acetylcholine), which upon binding to subjacent sensory neuron processes relay information about the filling status of the bladder to the central nervous system [10–13]. Yet, we have an incomplete understanding of the molecules that lower urinary tract tissues use to sense mechanical stimuli, the mechanotransduction pathways involved, or the contributions they make to normal and abnormal physiology. Thus, an important research goal is to identify lower urinary tract tissue-associated mechanosensors and assess their impact on voiding function.

Mechanosensors are biological force transducers that directly sense and respond to mechanical stimuli by undergoing conformational changes, ultimately triggering downstream mechanotransduction cascades [14–16]. Numerous classes of mechanosensors have been described including apical specializations (e.g., cilia), G-protein coupled receptors, elements of the cytoskeleton, proteins associated with cell-cell and cell-matrix junctions, and ion channels [14,17–26]. Examples of the latter include the PIEZO family channels PIEZO1 and PIEZO2, members of the two-pore $K^+$ channel family including KCNK2 (TREK-1), KCNK10 (TREK-2), KCNK4 (TRAAK), and orthologs of the plant osmosensitive OSCA1 channels identified in the mouse by Zhao *et al.* including TMEM63A and TMEM63B [27]. The latter are classified as high threshold, low conductance non-selective cation channels that are mechano- and osmo-sensitive [28–30].

TMEM63 family members are implicated in a variety of cellular functions and disease processes. In *Drosophila melanogaster*, a single TMEM63 ortholog is required for detection of humidity by Or42b neurons and detection of food texture by the md-L neurons that innervate sensory sensilla [31,32]. Also in *Drosophila,* as well as in the mouse neuroblastoma Neuro-2A cell line, orthologs of TMEM63A reportedly function as lysosomal mechanosensors [33]. *Tmem63a* may modulate chronic post-amputation pain [34], and mutations in *TMEM63A* are associated with transient hypo-myelination during infancy and hypomyelinating leukodystrophy [35–38]. TMEM63A in conjunction with TMEM63B function as mechanosensors in stretch-induced surfactant and ATP release by alveolar pneumocytes [39]. TMEM63B-dependent osmosensation is required for vertebrate hearing [28] and is also integral to thirst perception and detection of hyperosmolarity in the subfornical organ of the brain [40], the secretion of insulin [41], and release of thyroid hormone [42]. Gain-of-function variants of *TMEM63B* are associated with early-onset epilepsy, progressive brain damage, and hematological disorders [43]. An additional isoform, TMEM63C, has been described and is implicated in hereditary spastic paraplegia as well as defects in kidney podocyte function [44,45]; however, it does not function as a mechanosensitive

channel when reconstituted in liposomes [29]. What TMEM63C is doing in these contexts is unknown and awaits further exploration.

Our studies reveal that the urothelium and dorsal root ganglia (DRG; comprised of sensory neurons and accessory cells) are sites of TMEM6B/*Tmem63b* expression. To test the hypothesis that TMEM63B is integral to urothelial mechanotransduction, we generated conditional urothelial *Tmem63b* knockout (KO) mice and assessed their voiding behavior using video-monitored void spot assays. However, these mice had no observable voiding phenotype. Likewise, conditional sensory nerve *Tmem63b* KO mice exhibited no voiding defects. We discuss the implications of these findings, including the possibility that *Tmem63a,* which is also expressed by the urothelium and sensory neurons, compensates for loss of *Tmem63b* expression in these tissues.

## Materials and methods

### Animals and PCR genotyping

*Tmem63b*$^{HA-fl/HA-fl}$ mice, previously described by Du et al. [28], were generated by using CRISPR to insert nucleotides that encode an N-terminal HA tag (and 3X FLAG tag) after the 20-amino acid signal sequence. LoxP sites were also inserted into introns 1 and 4. The mice were backcrossed and maintained on a C57BL/6 background. Wild-type C57BL/6J mice, which we refer to as *Tmem63b*$^{+/+}$ mice in this manuscript, were obtained from the Jackson Laboratory (strain 000664; Bar Harbor, ME) and used as controls in some of our studies. *Avil*$^{Cre+/-}$ mice and *Upk2*$^{Cre+/-}$ mice were obtained from Jackson Laboratory (strains 032536 and 029281, respectively) [46,47]. As described previously [48], conditional urothelial *Tmem63b* KO mice were generated by crossing *Tmem63b*$^{HA-fl/HA-fl}$ mice with *Tmem63b*$^{HA-fl/HA-fl}$;*Upk2*$^{Cre+/-}$ mice. Conditional sensory neuron *Tmem63b* KO mice were generated by crossing *Tmem63b*$^{HA-fl/HA-fl}$ mice with *Tmem63b*$^{HA-fl/HA-fl}$;*Avil*$^{Cre+/-}$ mice. Expression of *Upk2* in all cell layers of the mouse bladder urothelium was confirmed by mating *Upk2*$^{Cre+/-}$ mice with Ai9 reporter mice (Jackson Laboratory strain 007909), which express CAG promoter-driven tandem-dimer-Tomato red fluorescent protein after Cre-mediated removal of a stop codon. Mice were housed in standard solid-bottom caging with up to five females/cage and up to four non-breeding males/cage. Animals were provided with paper blocks to generate bedding/nesting sites. Pregnant females were housed separately, as were male breeders. Mice were housed under a 12-hour day/night cycle and were fed standard mouse chow (Labdiet 5P76, irradiated; Purina, Wayne County, IN) and given water *ad libitum*. Mice were harem bred, and breeding males were housed one/cage. Genotyping for *Tmem63b*$^{HA-fl/HA-fl}$ was performed as previously described [28,48], and genotyping for *Upk2*$^{Cre+/-}$ and *Avil*$^{Cre+/-}$ was assessed using the primers and protocols provided on the Jackson Laboratory website. Experiments were performed with females and males between 9- and 24-weeks old.

Animal studies were performed in accordance with relevant guidelines/regulations of the Public Health Service Policy on Humane Care and Use of Laboratory Animals and the Animal Welfare Act, and under the approval of the University of Pittsburgh Institutional Animal Care and Use Committee (protocol number 24044849). Mice were euthanized by $CO_2$ asphyxiation, followed by thoracotomy as a secondary method.

### Reagents, antibodies, and fluorescent probes

Unless specified otherwise, all chemicals were obtained from Sigma-Aldrich (St Louis, MO). Rabbit monoclonal antibody (C29F4) to the HA epitope was purchased from Cell Signaling Technology (Danvers, MA; catalog number 3724S). Rabbit anti-cholera toxin B subunit antibody was from Novus (Centennial, CO; catalog number NB100−63607). The following secondary antibodies were purchased from Jackson ImmunoResearch Laboratories (West Grove, PA): donkey anti-rabbit-Alexa Fluor 488 (catalog number 711-545-152), goat anti-rabbit-Alexa Fluor 488 Fab fragments (catalog number 111-547-003), and goat anti-rabbit-CY3 (catalog number 111-165-144). Alexa Fluor 488- or rhodamine-labeled phalloidin (catalog numbers A12379 and R415, respectively) and DAPI (catalog number 62248) were obtained from ThermoFisher Scientific (Grand Island, NY).

## PCR analysis

Freshly excised mouse bladders were rinsed with Kreb's buffer (110mM NaCl, 25mM NaHCO$_3$, 5.8mM KCl, 1.2mM MgSO$_4$, 4.8mM KH$_2$PO$_4$, 11mM glucose, 2mM CaCl$_2$, gassed with 5% v/v CO$_2$). To isolate total RNA from bladders, the organ was cut into small pieces, weighed, and then placed in lysis/binding buffer at the ratio specified by the manufacturer (RNAqueous-4PCR Kit; Invitrogen, Waltham, MA). The included pestle was used to disrupt the tissue, and total RNA was isolated using the manufacturer's protocol. To enrich for urothelial RNA, fine forceps were used to invert the excised bladder onto the pointed end of a 200-µl yellow pipette tip, trimmed by 5mm with a scalpel, that was positioned next to the dome of the bladder. The inverted bladder was placed in 150µl of the above lysis/binding buffer for 30s and total RNA was isolated per the manufacturer's protocol. An AccuSript PfuUltra II RT-PCR kit (Agilent, Santa Clara, CA) was used to generate cDNAs from the isolated RNA using random primers and following the manufacturer's protocol. The following primers were used to identify gene expression by PCR:

*Tmem63a*-**FWR** TGAAAACGAGCTGGGATGCT

*Tmem63a*-**REV** CCTTTCTGGCTTCTCTGGGG

*Tmem63b*-**FWR** CTGCGCCTAGGGGAGGAT

*Tmem63b*-**REV** CGGAGGTGAGACGCTCATAC

*Tmem63c*-**FWR** GAGAGCAAGTTCCTGTGGCT

*Tmem63c*-**REV** GCTCCGCATCTACCTCCTTG

PCR reactions were performed using the KAPA HiFi polymerase kit (Roche Sequencing Solutions, Pleasanton, CA) and a T100 Thermal Cycler (BioRad, Hercules, CA) using the following protocol: initial denaturation at 94°C for 2min, followed by 39 cycles of 94°C for 20s, 60°C for 15s, 72°C for 15s, and a final step of 72°C for 2min prior to holding at 4°C. Amplicons were resolved using 2% w/v agarose gels.

## Fresh-frozen sample preparation

Mice were euthanized by CO$_2$ asphyxiation, their abdominal hair moistened using 70% v/v ethanol, and a caudal midline abdominal incision made using a sterile scalpel. The bladder was exteriorized and released from the body by cutting through its neck region just above the urethral sphincter using sharp tissue scissors. The bladder was rinsed in Kreb's buffer, plunged in Optimal Cutting Temperature (OCT) solution (Tissue-Tek, Sakura Finetek, Torrance, CA) to remove excess buffer, and then placed dome-side down in cryomolds (15 x 15 x 5mm; Fisher Scientific) filled with OCT. The samples were flash frozen by placing the cryomold on metal plate cooled with liquid nitrogen and stored at −80°C in sealed plastic bags. Cryosections were cut using a Leica Microsystems CM1950 cryostat (Buffalo Grove, IL; 8–12 µm sections; −20°C chamber and −18°C knife temperatures), collected on Superfrost Plus glass slides (ThermoFisher Scientific, Pittsburgh, PA), and held within the cryochamber at −20°C for 30min (to promote adherence of the section to the slide) prior to fixation/staining. Alternatively, slides were stored at −80°C.

## Detection and quantitation of gene expression by RNAscope and by BaseScope

RNAscope probes and associated multiplex RNAscope and Basescope kits were obtained from Advanced Cell Diagnostics (ACD; a Biotechne brand; Newark, CA). The following ACD probes were used for RNAScope analysis: *Tmem63a*-channel 1 (catalog number 431521), *Tmem63b*-channel 2 (catalog number 431531-C2), *Krt8*-channel 3 (catalog number 424521-C3), and B*acillus subtilis dapB* (which is detected in channels 1–3; catalog number 320871). *Krt8,* which is expressed by all urothelial cell layers, served as a positive control and *dapB* as a negative one. The ACD protocol for fresh-frozen tissue sections was

employed with the following changes. The frozen sections, prepared as described above, were fixed by adding cold (4°C), freshly prepared neutral buffered formalin (4% w/v paraformaldehyde dissolved in 29.04 mM $NaH_2PO_4 \cdot H_2O$ and 45.8 mM $Na_2HPO_4$, pH 7.4) and incubation for 15 min at room temperature. Protease IV treatment for 5 min at room temperature was employed. Opal reagent packs (Akoya Biosciences; Marlboro, MA) were used to fluorescently label RNAs: The Opal520 reagent pack for *Tmem63a*, Opal570 reagent pack for *Tmem63b*, and Opal690 reagent pack for *Krt8*. The Opal reagents were dissolved in 75µl of the provided DMSO, stored at 4°C, and used at a 1:1000 (Opal520/Opal570) or 1:3000 (Opal690) dilution during the RNAscope labeling protocol. Slides incubated with positive or negative control probes were run side-by-side and treated identically. Labeled tissue was mounted using ProLong Gold antifade mountant (ThermoFisher), cured overnight at room temperature in the dark, and the slides subsequently stored at 4°C prior to viewing and image acquisition. Quantification of *Tmem63a* and *Tmem63b* expression in the urothelium was performed as follows. For each of four bladders, images of six random sections, labeled with *Tmem63a*, *Tmem63b*, and *Krt8* RNAscope probes, were acquired using confocal microscopy (40X objective, 0.75 zoom, 2048 x 2048 image; see methods below for additional description of confocal microscopy). For each resulting image, urothelial-associated nuclei from all three urothelial cell layers were numbered and a random number table was used to identify five of these cells for further quantitation of *Tmem63a* and *Tmem63b* signal dots/urothelial cell nucleus. The number of urothelial cell nuclei that co-expressed *Tmem63a* and *Tmem63b* was also quantified. Results were tabulated in an Excel spreadsheet and like values were averaged per section and then per bladder.

To detect *Upk2^Cre*-mediated recombination of *Tmem63b^HA-fl/HA-fl*, a custom BaseScope probe called BA-Mm-Tmem63b-3zz-st-C1 was generated (catalog number 1129311-C). This probe was designed to detect the presence of a 218 bp fragment encoding exons 2–4 prior to Cre-mediated excision. The ACD BaseScope protocol was followed using fresh frozen tissue that was fixed and protease treated as described above. Positive and negative BaseScope control probes (catalog number 322976) were run concurrently. Images were collected using an HC PLAN APO 10X objective (N.A. 0.40) or an HCX PLAN APO 40X oil objective (N.A. 1.25) attached to a DM6000B widefield microscope (Leica Microsystems, Buffalo Grove, IL) outfitted with a Gryphax Prokyon digital camera (Jenoptik, Jupiter, FL), and using an Apple (Cupertino, CA) iMac computer running Gryphax software (Jenoptik). Images (1920 x 1200 pixels) were saved in TIFF file format. To quantify *Upk2-Cre*-dependent recombination, we opened random images of BaseScope-labeled tissues in Fiji (ImageJ2) and then used the freehand tool to mark the region of interest (ROI; i.e., urothelium). The area of the ROI was recorded, and the Cell Counter plugin was used to count the number of positive *Tmem63b* Basescope dots in this ROI. The number of dots per $mm^2$ of urothelial area were calculated and recorded in a Microsoft (Redmond, WA) Excel spreadsheet. For figures, images were contrast corrected in Adobe (San Jose, CA) Photoshop 2025 and composite images generated in Adobe Illustrator 2025.

## Confocal microscopy and image processing

Images were captured by confocal microscopy using either a Leica HCX PL APO 20X, 0.75 NA dry objective or a Leica HCX PL APO CS 40X, 1.25 NA oil objective (attached to a Leica DMI8 microscope) and the appropriate laser lines of a Leica Microsystems SP8 Stellaris confocal system outfitted with a 405-laser diode and a white-light laser. The signal from the Power HyD detectors was optimized using the Q-LUT option, and 8-bit images collected at 400–600 Hz using 3-line averages. Crosstalk between channels was prevented by use of spectral detection coupled with sequential scanning. Stacks of images (1024 x 1024, 8-bit) were collected using system-optimized parameters for the Z-axis. Images were processed using the 3D visualization option in Bitplane Imaris (Boston, MA) and exported as TIFF files. If necessary, the contrast of the images was corrected in Photoshop CC2025, and composite images prepared in Adobe Illustrator CC2025.

## Labeling afferent neurons with cholera toxin (Ctx) and isolation of DRGs

*Tmem63b^HA-fl/HA-fl* mice were anesthetized using isoflurane (3% v/v), the abdomen sterilized with povidone iodine, a caudal midline incision was made using a scalpel, and the bladder was exteriorized. Ctx beta subunit (MilliporeSigma; Burlington, MA; catalog number C9903), dissolved at 0.5% w/v in sterile 0.9% w/v saline, was injected into the bladder wall at

four sites (2 µl/injection site) using a Hamilton Company (Reno, NV) model 701 gastight microliter syringe outfitted with a 33g needle (Hamilton catalog number 7803−15). The abdominal incision was closed in layers using 5.0 polydioxanone absorbable monofilament surgical sutures (AD Surgical, Sunnyvale, CA, USA). Ketoprofen (5 mg/kg; Zoetis, Parsippany, NJ, USA) was administered subcutaneously to alleviate pain and ampicillin (100 mg/kg; Eugia, Hightstown, NJ, USA) was given to prevent infections. DRG from L4-L5 or those at L6-S2 were harvested 7 days later using our previously described protocols [49,50]. The DRGs from three mice were pooled, frozen in OCT compound, and sectioned as described above for bladder preparations.

### Detection of HA-TMEM63B and Ctx in tissues by immunofluorescence

Detection of HA-TMEM63B by immunofluorescence was performed as described previously [48]. Frozen sections of bladder and DRG (prepared as described above) were fixed by adding cold (4˚C), freshly prepared neutral buffered formalin and incubation for 10 min at room temperature. The tissue was rinsed with phosphate buffered saline (PBS; 137mM NaCl, 2.7mM KCl, 8.1mM $Na_2HPO_4$, and 1.5 mM $KH_2PO_4$, pH 7.4) and unreacted fixative was quenched by incubating the tissue slices for 10 min at room temperature with Quench Buffer (75 mM $NH_4Cl$ and 20 mM glycine, pH 8.0 dissolved in PBS, containing 0.1% v/v Triton X-100). The tissue was then rinsed 3 times with PBS followed by incubation in Block Solution (PBS containing 0.6% v/v fish skin gelatin and 0.05% w/v saponin) containing 10% v/v donkey serum for 60 min at room temperature in a humid chamber. The Block Solution was aspirated and replaced with primary antibodies diluted in Block Solution and incubated for 1 h at room temperature (or overnight at 4˚ C) in a humid chamber. The slides were washed 3-times quickly and 3 times for 3 min with Block Solution, and then incubated with fluorophore-labeled secondary antibodies, diluted in Block Solution, for 1 h at room temperature. DAPI (1:1000) and rhodamine-phalloidin (1:200) were included during the secondary antibody incubation. The labeled tissues were then rinsed 3-times quickly and 3 times for 5 min with Block Solution, rinsed with PBS, and then postfixed in neutral buffered formalin for 5–10 min at RT. The slides were rinsed with PBS, excess liquid aspirated, and a drop of SlowFade Diamond Antifade was placed on the tissue. Borosilicate coverslips (#1.5, 0.17 mm thickness, 24 x 50 mm; ThermoFisher) were placed above the drop of mounting medium, excess mounting medium was removed by aspiration, the edges of the coverslip were sealed with clear nail polish, and after the nail polish dried the slides were stored at −20 °C until image acquisition was performed. Control incubations lacked primary antibodies or secondary antibodies.

When simultaneously immunolocalizing Ctx and HA-TMEM63B, we used two rabbit primary antibodies. Following tissue fixation in neutral buffered formalin, the tissue was rinsed with PBS, then with Quench Solution, and subsequently incubated in Block Solution containing 5% v/v goat serum for 1 h at room temperature. The tissue was washed 3 times 3 min with Block Solution and then incubated with first primary antibody, rabbit anti-HA (diluted 1:100 in Block Solution), for 1 h at room temperature. The tissue was then washed 3 times 3 min with Block Solution and incubated for 1 h at room temperature with the first secondary antibody (goat anti-rabbit-Alexa Fluor 488 Fab fragments) diluted 1:500 in Block Solution. The tissue was then washed for 3 times 5 min with PBS. The antibodies were fixed using neutral buffered formalin for 10 min at room temperature, rinsed with PBS and then Quench buffer. The tissue was incubated with Block Solution for 5 min and then reacted with the second primary antibody (rabbit anti-Ctx B subunit) diluted 1:500 in Block Solution for 60 min. The tissue was washed 3 times 5 min with Block Solution and then incubated with the second secondary antibody (CY3-conjugated goat-anti-rabbit antibody; 1:3000 dilution) for 60 min at room temperature. The samples were then washed 3 times 5 min with Block Solution, rinsed with PBS, and then post-fixed and mounted as described above. Control incubations included the following: (1) omission of the first primary antibody; (2) omission of the second primary antibody; (3) omission of the first primary antibody, the first secondary antibody, and the second primary antibody; (4) omission of the first primary antibody, the second primary antibody, and the second secondary antibody.

## Video-monitored void spot analysis

To assess voiding behavior in awake, freely moving mice, we used custom-built void-spot chambers outfitted with video monitoring, and the data analyses described previously [2,51]. In brief, these chambers included an upper compartment that housed an individual mouse with continuous access to food and water (in the form of Hydrogel; ClearH2O, Westbrook, ME) and an igloo-shaped sleeping chamber. The floor of the upper chamber was lined with chromatography/blotting paper and illuminated from below by UV lights housed in a lower chamber. The animals and paper were monitored by wide-angle "webcam" cameras positioned above the upper chamber and another mounted at the base of the lower compartment. The incorporation of real-time video monitoring allowed us to follow mouse activity over extended periods of time while overcoming the difficulties of distinguishing overlapping voiding spots, a shortfall of standard void-spot assays [52]. The mice were routinely housed in a facility with 12-hour light-dark cycles, with 7:00AM being zeitgeber time (ZT)=0 (start of light cycle). To analyze their voiding behavior in the dark phase, the mice were placed in the upper chamber between 17:00–18:00 h (ZT10–11), and analysis of void spots was performed from 00:00–6:00 (ZT=17–23). The extended period of acclimatization reflected access limitations to the facility after 7:00 PM. For analysis during the light phase, the mice were placed in the chamber between 10:00–11:00 (ZT=3–4), allowed to acclimatize for one hour, and analysis performed during the subsequent 6-h time window. Video was captured at 1 frame per second with a 1920×1080 pixel resolution using an Apple M3 Mac mini-computer running SecuritySpy software (BenSoftware.com). The movies were saved in.m4v format and viewed on an Apple Intel i9 iMac computer using Quicktime (Apple) software. Calibration curves, made by spotting mouse urine (2 µl – 750 µl) on the paper, were used to calculate the volume/void spot. As previously [2,51], spots were categorized as primary void spots (PVS; those ≥ 20 µl) or secondary void spots (SVS; < 20 µl). Because of the possibility of dribbling and trailing of urine spots, all void spots occurring within a 60s period-of-time were treated as a single voiding event. The parameters measured in our analysis were number of PVS, average PVS volume per void, total PVS volume, number of SVS, total SVS volume, and frequency of voiding events ([PVS+SVS]/h). Animals that chewed or damaged the paper prior to or during the time window of analysis were excluded from the study.

## Cyclophosphamide treatment and analysis

An acute (single dose) cyclophosphamide (CYP) model was employed. A 25 mg/ml stock of CYP, dissolved in sterile 0.9% w/v saline, was prepared daily and filtered through a sterile 0.22 µm syringe filter. In unpaired experiments, mice were injected once intraperitoneally with CYP (150 mg/kg; i.e., 6 µl of 25 mg/ml stock injected per g mouse weight) or with vehicle (0.9% w/v saline) at 10:00–11:00 (ZT=3–4), transferred to the void-spot chamber at 17:00–18:00 h (ZT10–11), and void-spot analysis performed in the subsequent 0:00–6:00 (ZT=17–23) time frame. The effects of CYP (i.e., increased urination) in female mice were observable within one hour of injection. In paired experiments, mice were injected with saline at 10:00–11:00 (ZT=3–4), transferred to the void spot chamber at 17:00–18:00 h (ZT10–11), and void spot analysis performed in the subsequent 0:00–6:00 (ZT=17–23) time window. Next, the animals were injected with CYP at 10:00–11:00 (ZT=3–4) and void-spot analysis performed from 0:00–6:00 (ZT=17–23).

## Statistical analysis

Data are expressed as mean±SEM (*n*), where *n* equals data from an individual mouse. Parametric or nonparametric tests were employed as appropriate. CYP experiments were analyzed using 2-way ANOVA. Values of $p \leq 0.05$ were considered statistically significant. Statistical comparisons were performed using GraphPad Prism 10 (GraphPad Software, San Diego, CA, USA).

## Results

### Expression of *Tmem63a* and *Tmem63b*/TMEM63B in the mouse bladder

An important site of mechanotransduction in the bladder wall is the urothelium [1,13]. We used RT-PCR to demonstrate that *Tmem63a* and *Tmem63b* were both expressed in the bladder and specifically in the urothelium (Fig 1A). In contrast,

we could not detect expression of *Tmem63c* in these tissues (Fig 1A). These observations fit well with transcriptomic studies that report expression of *Tmem63a* and *Tmem63b* in the cells that populate the bladder, but not *Tmem63c* [53]. RNAscope analysis (a variant of fluorescent in situ hybridization) revealed that *Tmem63a* expression was concentrated in the urothelium, but with scattered expression detected in the lamina propria and the muscularis externa (Fig 1B-C). *Tmem63b* expression was also concentrated in the urothelium, but with less signal apparent in suburothelial tissues (Fig 1B-C). Quantitation revealed equivalent expression of *Tmem63a* and *Tmem63b* (~ 6 signal dots/cell) in urothelial cells sampled across the three urothelial cell layers. In all cells sampled, both gene products were co-expressed (Fig 1D). Limited reaction product was detected in samples incubated with the *B. subtilis dapB* triple-negative control probe (Fig 1B-C). To confirm TMEM63B protein expression in the urothelium, we used a previously published reporter mouse (*Tmem63b*$^{HA-fl/HA-fl}$), which expresses an HA tagged version of the TMEM63B protein (which we refer to as HA-TMEM63B) in place of the endogenous protein [28]. Consistent with our RNAscope analysis, we observed that HA-TMEM63B was highly enriched in the urothelium and not the other tissues of the bladder wall (Fig 2A). Within the urothelium, the intensity of staining was polarized, with the greatest signal in the basal cell layer and decreasing amounts of signal as one progressed from intermediate cell layer to the outermost umbrella cell layer (see right-most panels of Fig 2D). This indicates that despite mRNA expression across the urothelium, protein expression is greatest in the basal cell layer. Whether this reflects differences in translation or protein turnover was not assessed further. As a control for these studies, we demonstrated that HA-TMEM63B was not detected in the urothelium of wild-type (*Tmem63b*$^{+/+}$) mice processed identically (Fig 2A). A reporter mouse expressing a tagged version of *Tmem63a* was not available, so similar studies could not be performed for this gene product.

## Generation of conditional urothelial specific Tmem63b KO mice

Given the preponderance of *Tmem63b*/TMEM63B expression in the urothelium, combined with the availability of *Tmem63b*$^{HA-fl/HA-fl}$ mice, we next set out to make conditional urothelial *Tmem63b* KO mice. Within the bladder, *Upk2*$^{Cre}$ mice are reported to drive selective expression of *Cre* in the mouse urothelium [46,54], and we confirmed that that *Upk2*$^{Cre}$ mice employed in our studies drive Cre expression in all three strata of the urothelium (i.e., umbrella cell layer, intermediate cell layers, and basal cell layer) (Fig 2B). To confirm knockout of *Tmem63b* in the urothelium, we used an *in situ* hybridization approach (BaseScope), which confirmed almost complete Cre-mediated excision of exons 2–4 in the urothelium of conditional urothelial *Tmem63b* KO mice (*Tmem63b*$^{HA-fl/HA-fl}$;*Upk2*$^{Cre+/-}$) versus control ones (*Tmem63b*$^{HA-fl/HA-fl}$;*Upk2*$^{Cre-/-}$) (Fig 2C). Furthermore, we observed a loss of urothelial HA-TMEM63B protein expression in conditional urothelial KO mice when compared to control ones (Fig 2D).

## Voiding behavior is not affected in conditional urothelial *Tmem63b* KO mice

A useful screening tool for detecting defects in lower urinary tract function, including alterations in mechanotransduction, is the void spot assay [2,3,55–57]. We recently developed a video-monitored version of this approach and used it to assess voiding behavior in male and female control and conditional urothelial *Tmem63b* KO mice during their active (dark) and resting (light) phases [2,51]. As described previously, we binned voids into primary voids spots (PVS; defined as ≥ 20 µl) or secondary void spots (SVS; defined as < 20 µl) [2,51]. The former are associated with normal voiding behavior, while the latter are somewhat more prevalent in dominant males and are significantly increased in animals with an overactive bladder phenotype [52,58–60]. However, we did not observe a significant difference in PVS parameters (PVS number, average PVS volume, and total PVS volume), SVS parameters (SVS number and total SVS volume), or frequency when comparing control and conditional urothelial female *Tmem63b* KO mice (Fig 3A). A void spot analysis was also performed in conditional urothelial male *Tmem63b* KO mice and controls (Fig 3B). Again, there was no significant difference in measures of PVS number, average PVS, total PVS, SVS number, total SVS, or frequency in light or dark phases.

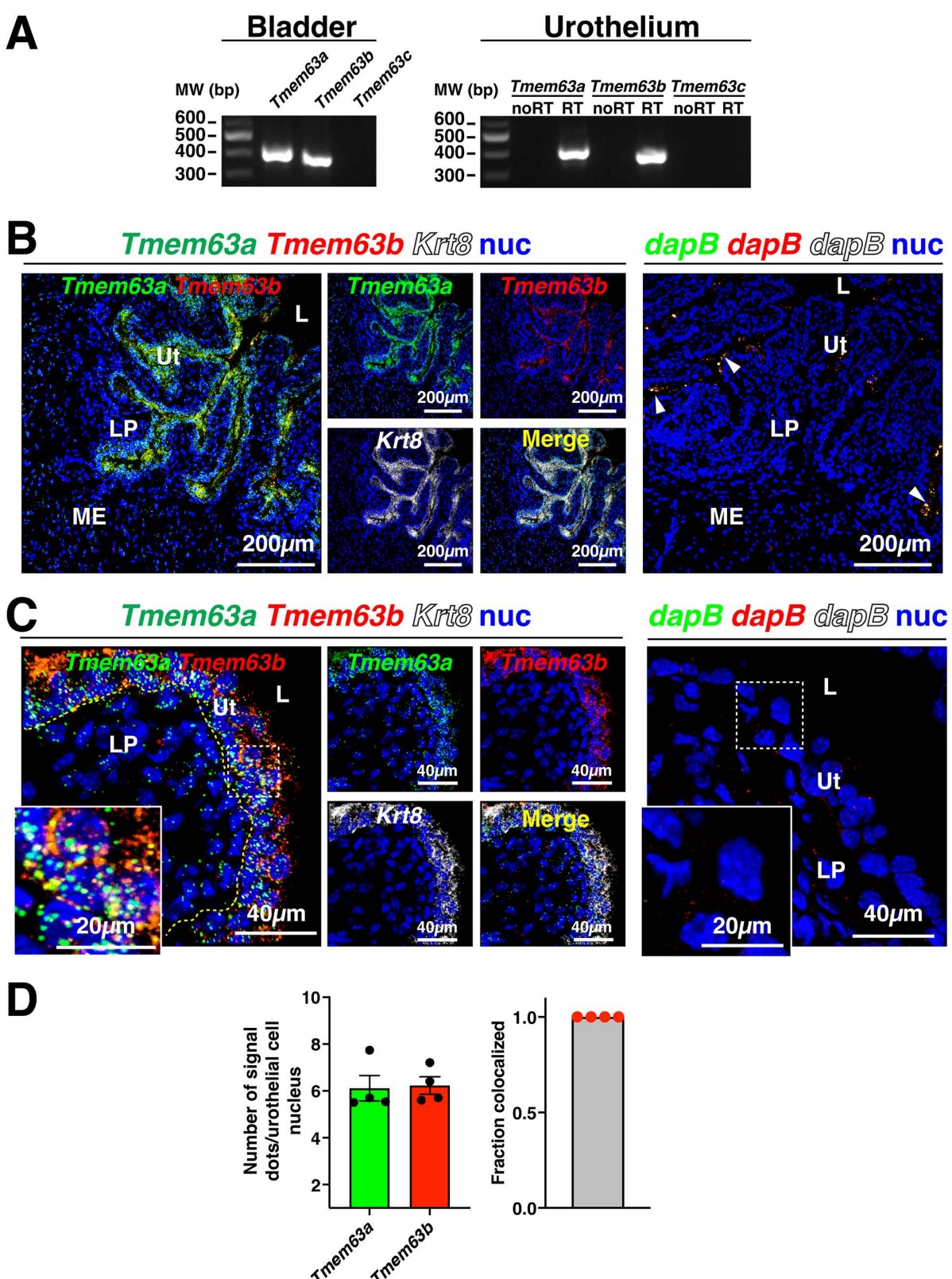

**Fig 1. Expression of *Tmem63a* and *Tmem63b* in the bladder urothelium.** (A) mRNA was isolated from whole bladder (left-most panel) or urothe-lium (right-most panel), reverse transcribed, and PCR used to detect the presence of the indicated gene product. (B-C) Expression and localization of

message for the indicated gene as assessed by RNAscope. Nuclei (nuc) were visualized using DAPI labeling. The *Krt8* probe, which labels all three layers of the urothelium, served as a positive control. The *dapB* probe serves as a negative control. Lower magnification overviews are provided in B and higher magnification views are included in C. The arrowheads in the right-most panel of B indicate umbrella cell-associated lysosomes, which variably exhibit autofluorescence. The boxed regions in C are magnified in the insets. (D) Left panel: *Tmem63a* and *Tmem63b* expression quantified as number of signal dots per urothelial cell nucleus. Right panel: fraction of sampled cell nuclei co-expressing both *Tmem63a* and *Tmem63b*. Data are mean ± SEM (n = 4 animals, two males and two females). *Legend:* L, lumen; LP, lamina propria; ME, muscularis externa; Ut, urothelium.

We next assessed whether a function for *Tmem63b* would be revealed if the urothelium was stressed. In this case, we performed void spot analysis in animals treated with cyclophosphamide (CYP), an anti-cancer and anti-inflammatory drug that is rapidly converted into the toxic metabolites phosphoramide mustard and acrolein, prior to excretion into urine [61–63]. Acute CYP treatment results in significant edema and swelling of the bladder lamina propria, disruption of the urothelium, and a hemorrhagic cystitis within hours of injection [64]. Relative to animal treated with vehicle (i.e., saline) alone, CYP-treated female mice exhibited all of the hallmarks of an overactive bladder phenotype including a significant increase in the number of SVS (Fig 4). These changes were accompanied by a significant decrease in number of PVS, average PVS volume, and total PVS volume. Despite these changes, there was no significant difference between the voiding behavior of control female mice versus conditional urothelial *Tmem63b* KO mice in response to saline or CYP treatment (Fig 4). We also assessed voiding behavior in CYP-treated male urothelial *Tmem63b* KO mice. In contrast to female mice, male ones were relatively resistant to CYP and the only significantly different parameter versus saline vehicle was SVS number. Again, there was no significant difference between male control mice versus conditional urothelial *Tmem63b* KO ones. On balance, we found no evidence that conditional urothelial *Tmem63b* KO mice exhibited altered voiding behavior under the conditions we tested.

## HA-TMEM63B is expressed in dorsal root ganglia neurons but loss of *Tmem63b* expression in DRGs does not impact voiding behavior

Additional sites of mechanotransduction in the bladder wall are the afferent processes that innervate the bladder wall and convey information about the mechanochemical status of bladder tissues to spinal cord-associated DRG neurons [3,65–73]. RNAscope analysis revealed co-expression of *Tmem63a* and *Tmem63b* in L6-S2 DRG (i.e., which include the sensory neurons that innervate the bladder) (Fig 5A) and immunofluorescence confirmed that a subset of these DRGs expressed HA-TMEM63B (Fig 5B). No HA signal was detected in the absence of primary antibody (Fig 5C). Retrograde tracing studies, employing cholera toxin beta subunit (Ctx) that was injected into the bladder wall, confirmed that HA-TMEM63B was expressed in the sensory neurons that innervated the urinary bladder (Fig 5B). As expected, none of the DRGs positioned at L4-L5 were labeled with Ctx (Fig 5B). However, these latter neurons expressed HA-TMEM63B, indicating that TMEM63B-positive DRG neurons likely innervate other regions of the body (in the case of L4-L5, this includes the lower back, hindlimbs, and some pelvic organs). To target *Tmem63b* expression in afferent neurons, including those innervating the bladder, we used *Avil^Cre* mice [47]. We confirmed that conditional sensory neuron *Tmem63b* KO mice (*Tmem63b^HA-fl/HA-fl*;*Avil^Cre+/-*) exhibited a large decrease in expression of HA-TMEM63B relative to control mice (*Tmem63b^HA-fl/HA-fl*;*Avil^Cre-/-*)(Fig 5C).

Both female and male control and conditional sensory neuron *Tmem63b* KO mice were subjected to void-spot analysis. Regardless of sex, or dark/light phase, we did not detect significant differences between control and conditional sensory neuron KO mice (Fig 6A-B). CYP is reported to result in sensory neuron hypersensitivity [50,74,75]. Thus, we also assessed voiding behavior in CYP-treated control and conditional sensory neuron KO mice. We were able to detect significant differences in the voiding parameters of CYP-treated female mice versus vehicle-treated ones, but this was not true of male mice (Fig 7). However, there was no significant difference in any of the measured parameters when comparing control mice *(Tmem63b^HA-fl/HA-fl*;*Avil^Cre-/-*) and conditional sensory neuron KO ones (*Tmem63b^HA-fl/HA-fl*;*Avil^Cre+/-*) of either sex (Fig 7).

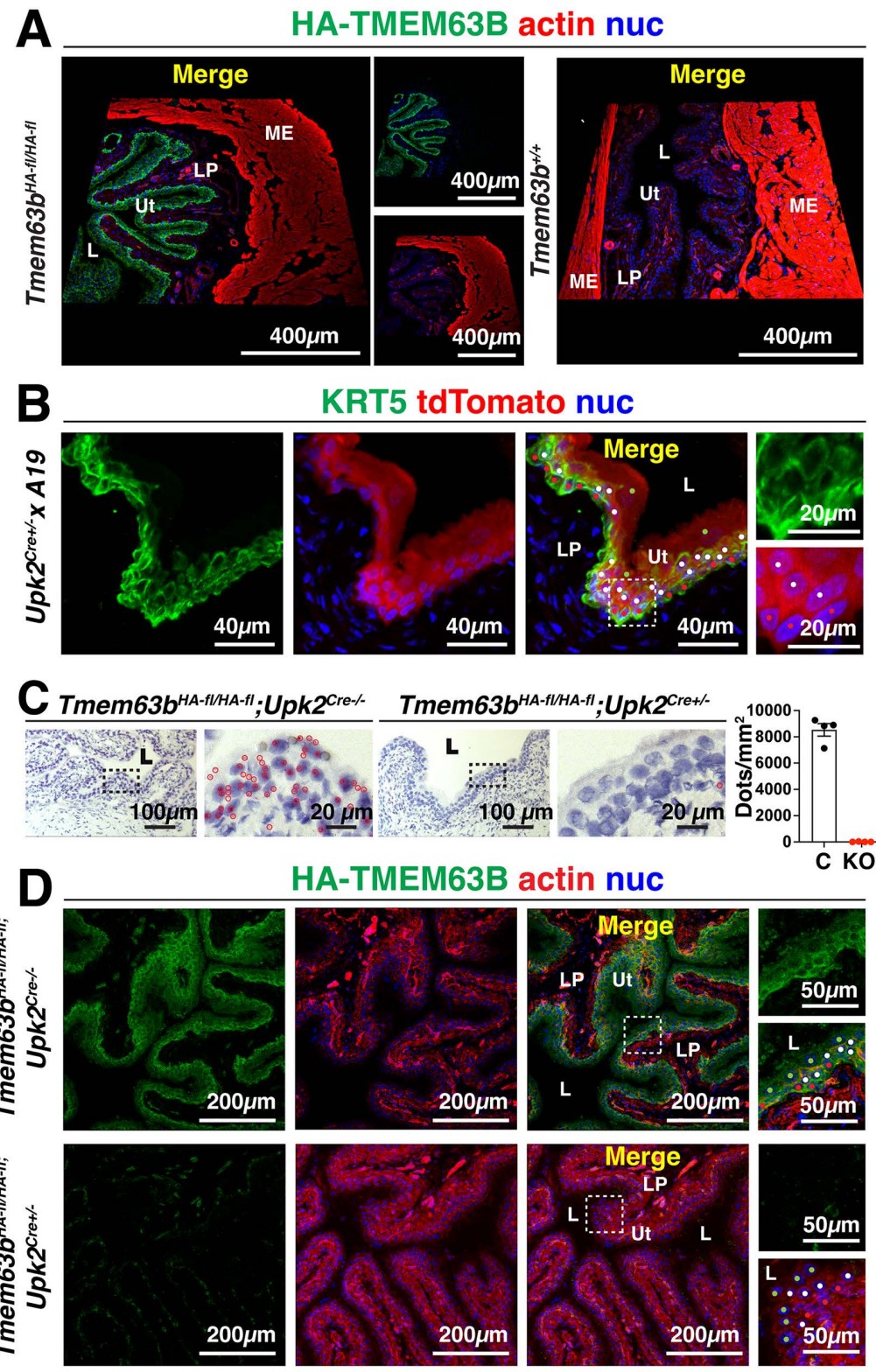

**Fig 2. Expression of HA-TMEM63B/*Tmem63b* in conditional urothelial *Tmem63b* KO mouse bladders.** (A) Localization of HA-TMEM63B in the bladders of *Tmem63b^HA-fl/HA-fl^* and wild-type *Tmem63B^+/+^* mice. (B) *Upk2^Cre+/-^* mice were crossed with Ai9 reporter mice and expression of tandem dimer

(td)Tomato in the KRT5-labeled urothelial cells was assessed by immunofluorescence. The boxed region is magnified in the panels to the right. (C) Use of BaseScope to assess *Upk2*-Cre-mediated recombination of *Tmem63b*$^{fl-HA/fl-HA}$ exons 2-4. Reaction products (dots), indicating expression of *Tmem63b* exons 2-4, are marked with red circles. In the panel to the right, the number of dots per mm$^2$ of urothelium was quantified in *Tmem63b*$^{HA-fl/HA-fl}$;*Upk2*$^{Cre-/-}$ (control; C) and *Tmem63b*$^{HA-fl/HA-fl}$;*Upk2*$^{Cre+/-}$ (conditional knockout; KO) mouse bladders. Data are mean±SEM (n=4). D. HA-TMEM63B expression in *Tmem63b*$^{HA-fl/HA-fl}$; *Upk2*$^{Cre-/-}$ and *Tmem63b*$^{HA-fl/HA-fl}$;*Upk2*$^{Cre+/-}$ mice. In panels A, B, and D nuclei (nuc) are labeled with DAPI. In panels B and D, red dots indicate basal cells, white dots indicate intermediate cells, and green dots indicate umbrella cells. *Legend:* L, lumen; LP, lamina propria; Ut, urothelium.

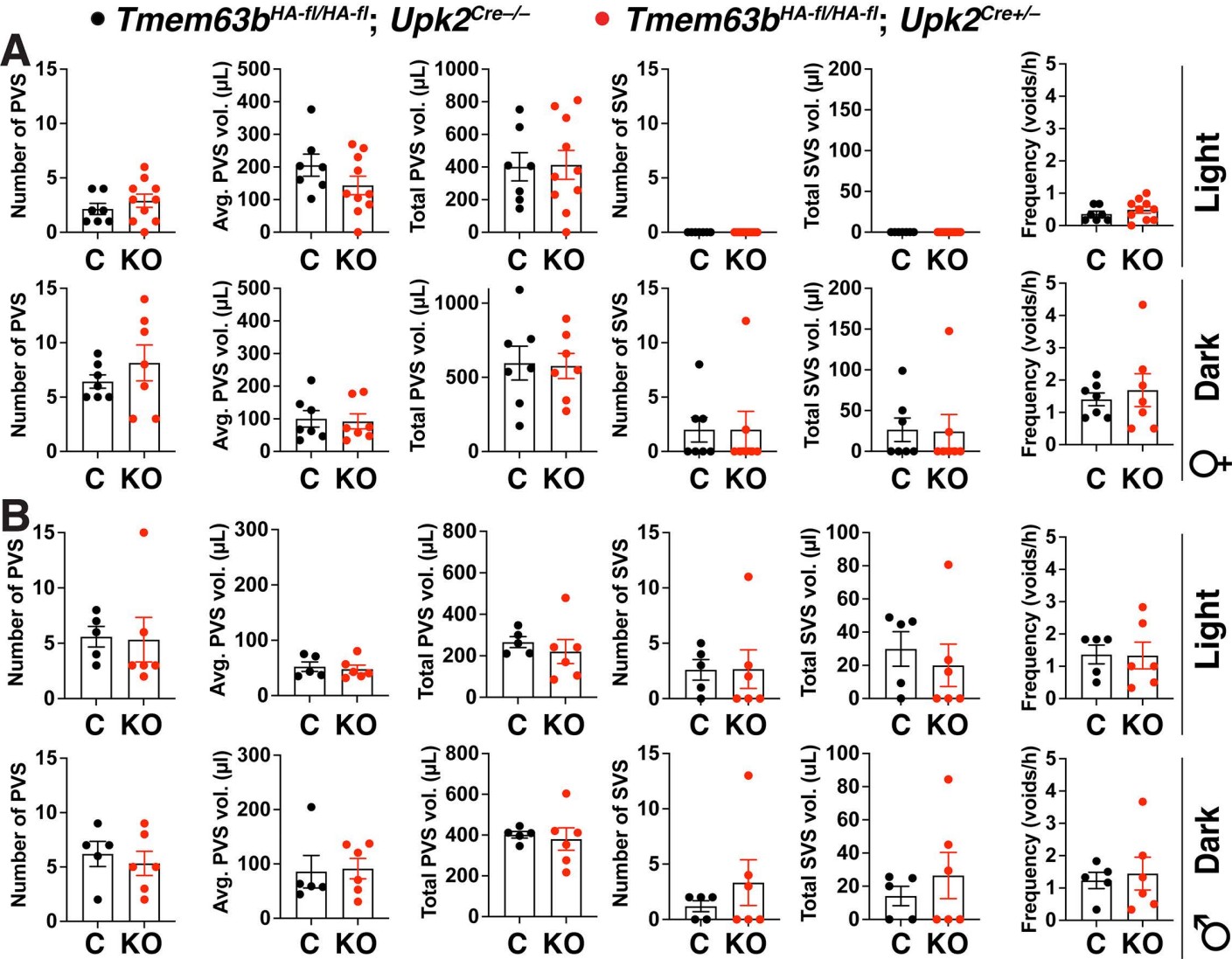

**Fig 3. Voiding behavior in control and conditional urothelial *Tmem63b* KO male and female mice during their light and dark phases.** (A-B) Video-monitored void spot analysis was performed in female and male *Tmem63b*$^{HA-fl/HA-fl}$;*Upk2*$^{Cre-/-}$ (control; C) and conditional urothelial *Tmem63b*$^{HA-fl/HA-fl}$;*Upk2*$^{Cre+/-}$ (knockout; KO) mice during the indicated light/dark phase. Measured parameters included: number of primary void spots (PVS), average number of PVS, total volume of PVS, number of small void spots (SVS), total volume of SVS, and frequency (total number of voids/h). Data, mean±SEM (n ≥ 5), were analyzed using Mann-Whitney tests. No significant differences were detected.

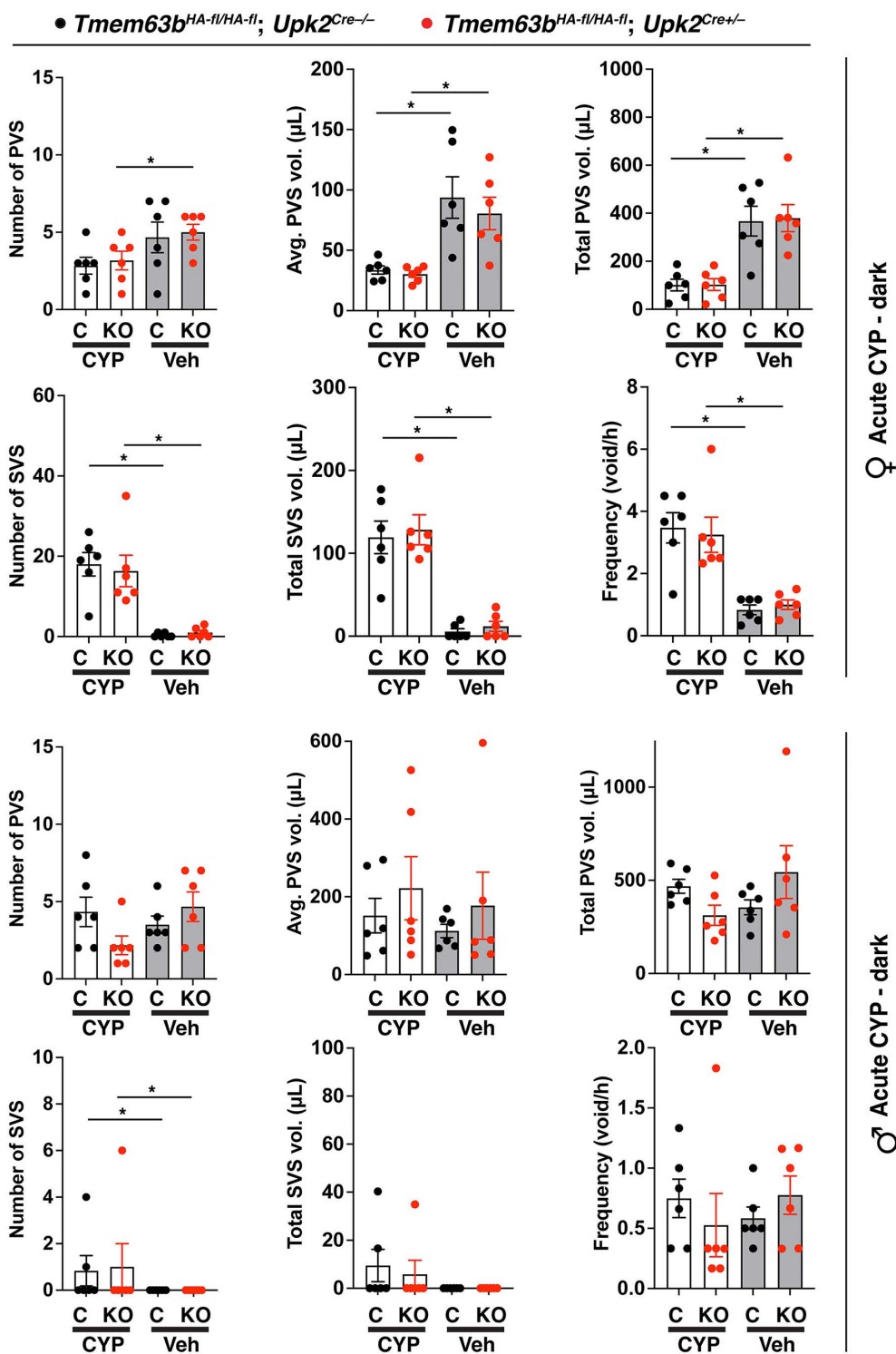

**Fig 4. Effects of acute cyclophosphamide treatment on voiding behavior in conditional urothelial *Tmem63b* KO mice.** Video-monitored void spot analysis was performed in female and male control *Tmem63b*^HA-fl/HA-fl^;*Upk2*^Cre-/-^ (C) and conditional urothelial *Tmem63b*^HA-fl/HA-fl^;*Upk2*^Cre+/-^ (KO) mice during their dark phase. Mice were subjected to VSA analysis after injection with vehicle (Veh; saline) and subsequently after injection with 150 mg/kg of cyclophosphamide (CYP). Data are mean±SEM (n = 6). Paired data were analyzed using two-way ANOVA with Šidák's multiple comparisons test. Significant differences (p < 0.05) are indicated with asterisks.

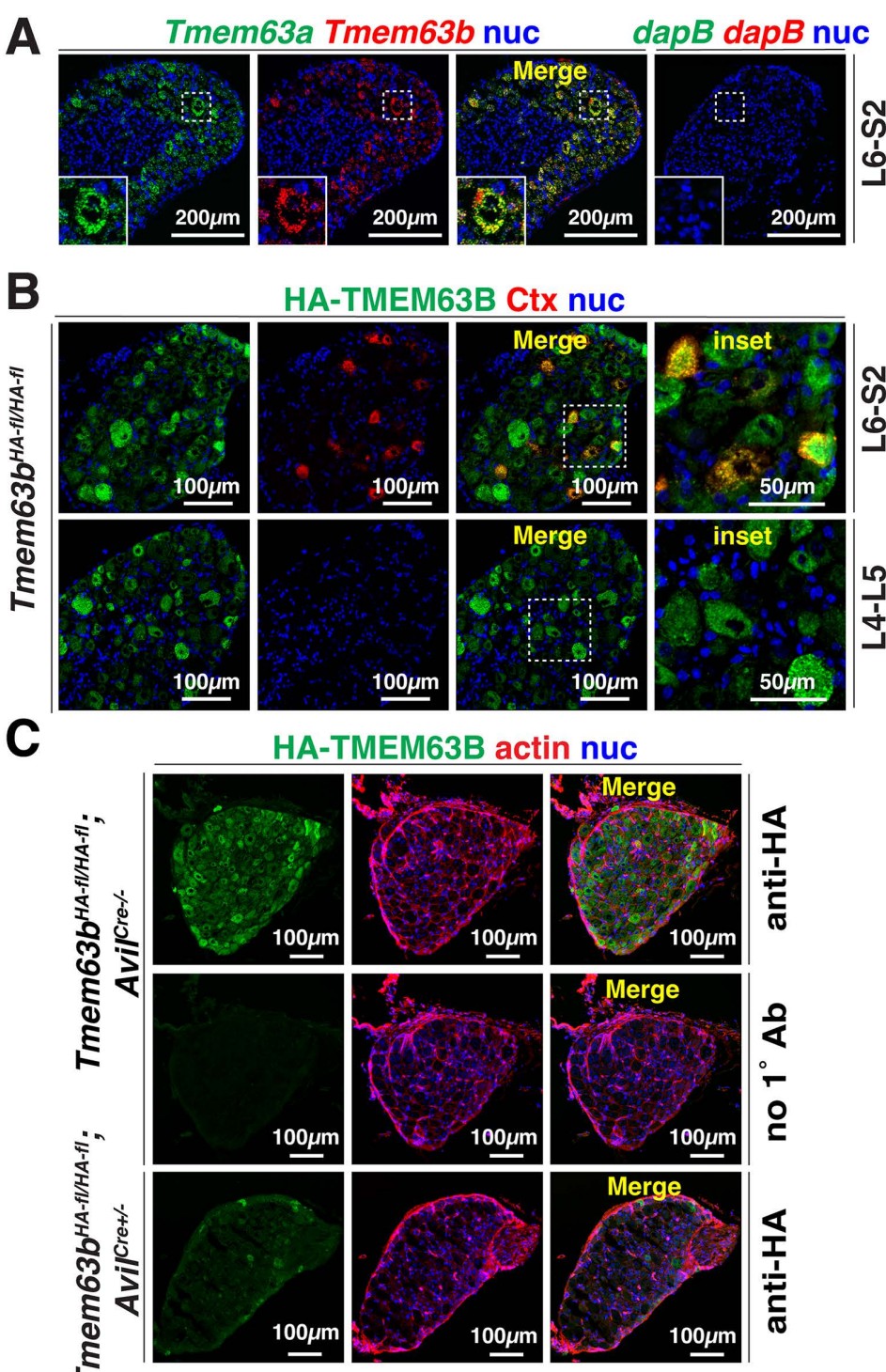

**Fig 5. Expression of *Tmem63a*, *Tmem63b,* and HA-TMEM63B in sensory neurons.** (A) Left-most panels: expression of *Tmem63a* and *Tmem63b* in L6-S2 DRG neurons. Right-most panel: DRG labeled with the *dapB* triple-negative control probe. The boxed regions are magnified in the insets. (B) HA-TMEM63B is expressed in cholera toxin (Ctx)-labeled bladder L6-S2 DRG neurons. HA-TMEM63B is also expressed in Ctx-negative L4-L5 DRG neurons. The boxed regions in the panels labeled "merge" are magnified in the right-most panels labeled "inset." (C) HA-TMEM63B expression in control mice (*Tmem63b^{HA-fl/HA-fl}*;*Avil^{Cre-/-}*) *a*nd conditional sensory neuron KO mice (*Tmem63b^{HA-fl/HA-fl}*;*Avil^{Cre+/-}*). A no primary antibody control is included in the middle panel of this figure. In A-C, nuclei (nuc) were labeled with DAPI.

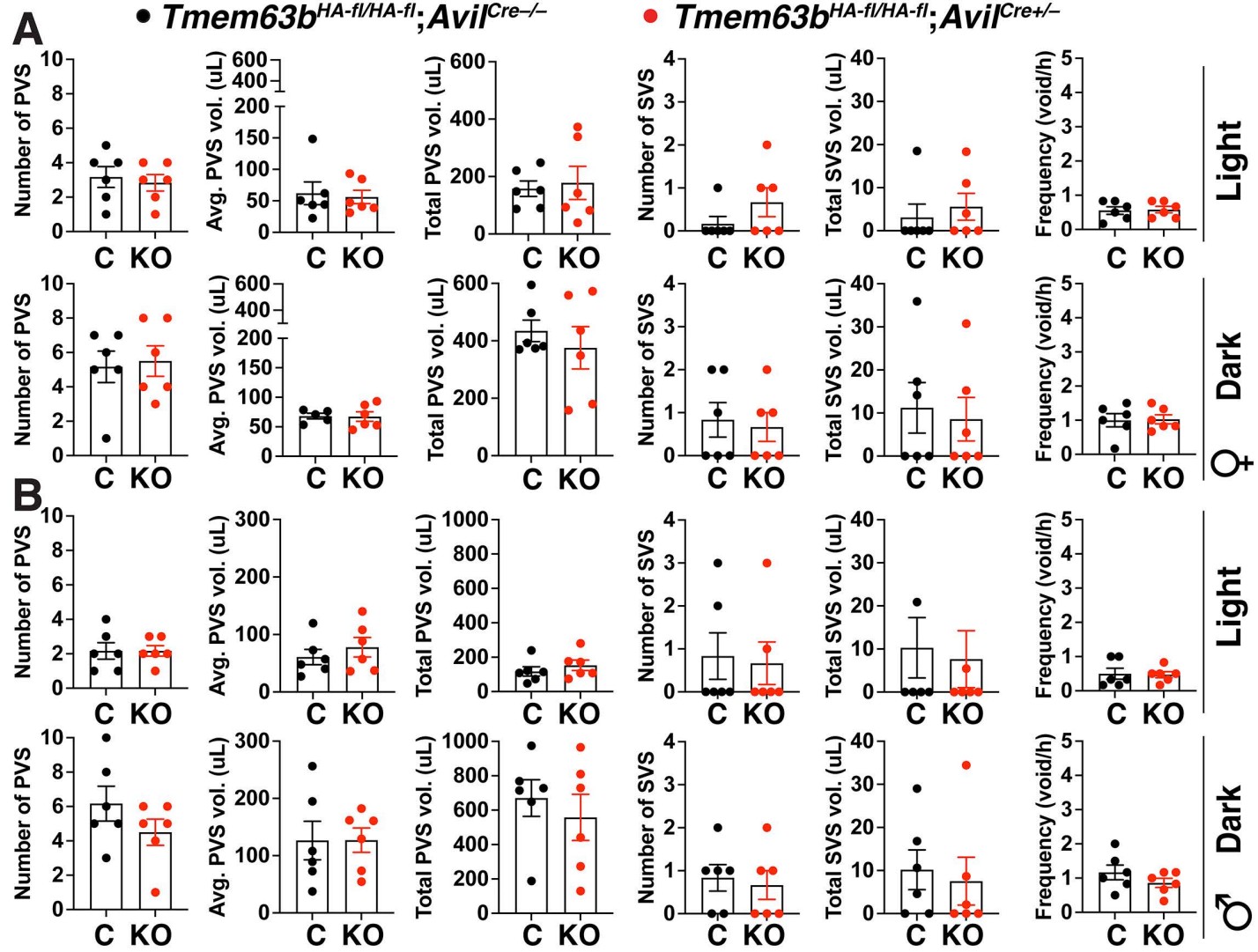

**Fig 6. Voiding behavior in control and conditional sensory neuron *Tmem63b* KO male and female mice during their light and dark phases.** (A-B) Video-monitored void spot analysis was performed in female and male control *Tmem63b^HA-fl/HA-fl^;Avil^Cre-/-^* (C) and conditional sensory neuron *Tmem63b^HA-fl/HA-fl^;Avil^Cre+/-^* KO mice during the indicated light/dark phase. Data are mean±SEM (n = 6). Data were analyzed using Mann-Whitney tests. No significant differences were detected.

## Discussion

Despite numerous reports that the urothelium and associated sensory neurons are mechanosensitive [1,13,76], the relevant mechanosensors and downstream mechanotransduction pathways involved in these pathways are only slowly coming into focus. Earlier evidence implicated *Itgb1* (integrin beta 1), *Trpv4*, and *Slc17a9* (vesicular nucleotide transporter) in urothelial mechanosensation [55–57]. Integrins have a well-established function in mechanotransduction [26,77], and conditional urothelial *Itgb1* KO mice exhibit elevated mucosal ATP release (serosal release was not determined), as well as bladder overactivity and incontinence when assessed in void-spot studies [55]. The exact mechanosensory pathways affected in these mice has not yet been defined, although it is worth noting that there is increasing evidence of cross-talk between PIEZO1 signaling (see discussion below) and focal adhesions [78]. Whole animal *Trpv4* KO mice exhibit

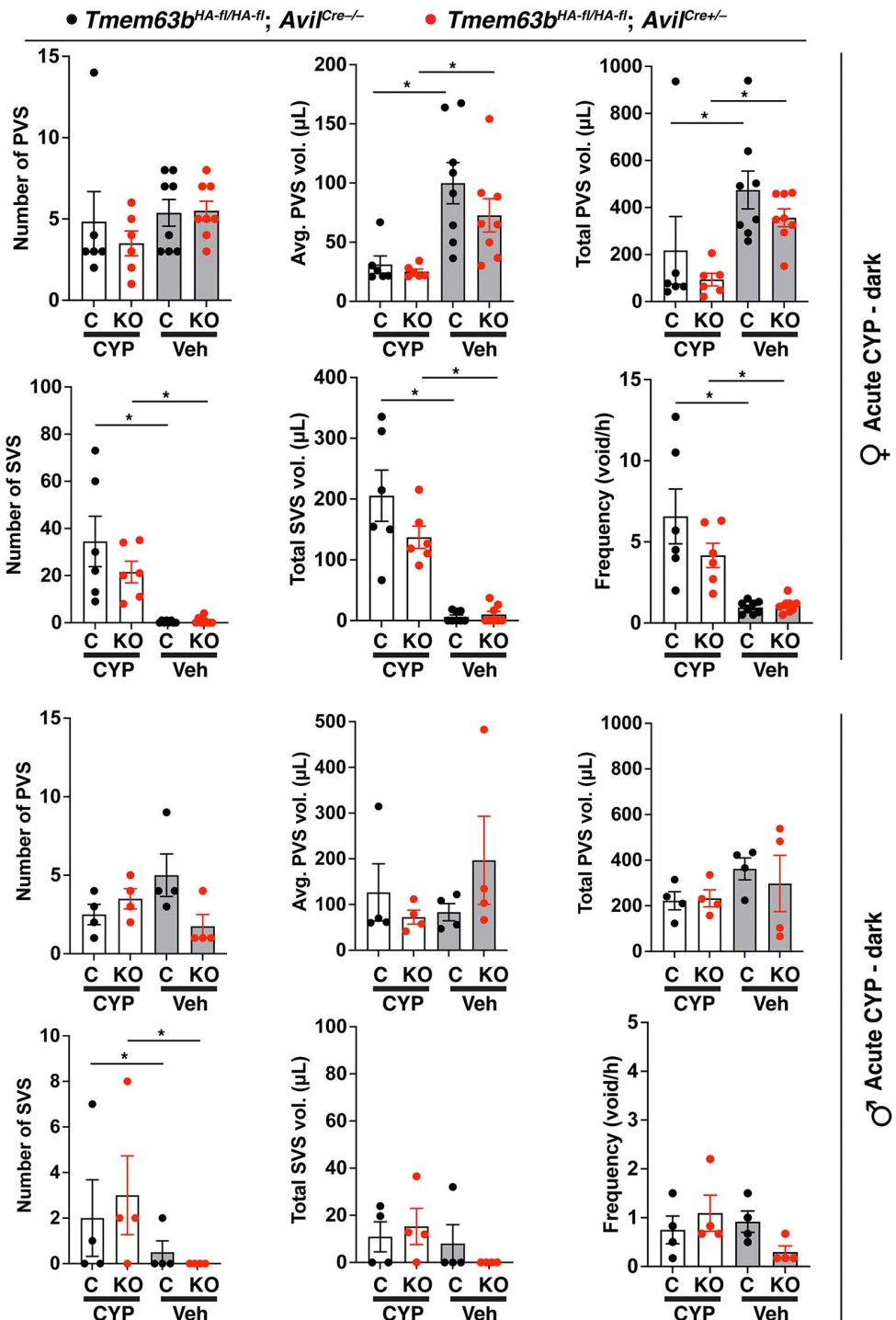

**Fig 7. Effects of acute cyclophosphamide treatment on voiding behavior in conditional sensory neuron *Tmem63b* KO mice.** Video-monitored void spot analysis was performed on female and male control *Tmem63b*^HA-fl/HA-fl^;*Avil*^Cre-/-^ (C) and conditional sensory neuron *Tmem63b*^HA-fl/HA-fl^;*Avil*^Cre+/-^ (KO) mice during their dark phase. In these experiments, female mice were injected with saline vehicle (Veh) or 150 mg/kg of cyclophosphamide (CYP) prior to VSA analysis. The resulting unpaired female data were analyzed using two-way ANOVA with Tukey's multiple comparisons test. Male mice were injected with saline vehicle (Veh) and VSA analysis undertaken. The same mice were subsequently injected with 150 mg/kg of cyclophosphamide (CYP) and then subjected to an additional round of VSA analysis. The resulting paired male data were analyzed using two-way ANOVA with Šidák's multiple comparisons test. All data are mean±SEM (n ≥ 4). Significant differences (p < 0.05) are indicated with asterisks.

decreased mucosal ATP release (again, serosal release was not measured) and increased numbers of void spots [56]. Parsing the affected tissue types is difficult as multiple cell types in the bladder wall express *Trpv4,* as do DRGs [53,79–81]. Furthermore, it remains controversial whether *Trpv4* functions as a mechanosensor or whether it is activated downstream of initiating mechanosensory events; however, this may depend on cell context [82,83]. Whole animal *Slc17a9* KO mice exhibit decreased mucosal ATP release (serosal release not measured) when the bladder is filled with small fluid volumes [57]. In addition, these mice exhibit decreased bladder compliance and have an increased tendency to void away from their corners when analyzed in void-spot assays. *Slc17a9* is expressed by the urothelium as well as sensory neurons [57,84]. Thus, the underlying target(s) of the KO phenotype is not yet clear.

More recent data implicate the well described mechanosensory proteins PIEZO1 and PIEZO2 in urothelial and afferent function [85]. In our hands, individual conditional urothelial *Piezo1* or *Piezo2* KO mice exhibit a limited bladder phenotype [2]. In contrast, double conditional urothelial *Piezo1/2* KO are characterized by an almost complete loss of urothelial serosal ATP release and altered bladder and voiding behavior that varies in a sex- and circadian rhythm-dependent manner [2]. Worth noting though, Marshall *et al.* report that conditional urothelial *Piezo2* KO mice exhibit altered bladder and sphincter function, but they did not measure the impact of *Piezo2* KO on urothelial mechanotransduction or voiding behavior [3]. One would predict that ablating urothelial pathways for mechanotransduction would lead to a hypoactive bladder. Instead, conditional urothelial *Itgb1KO mice, Trpv4* KO mice, and double *Piezo1/2* KO mice have an overactive bladder phenotype (characterized by large numbers of small voiding events) indicating that urothelial mechanotransduction may play a role in suppressing voiding function and not simply stimulating it [2]. Sensory neurons are also known to express *Piezo2* and Marshall *et al.* report that conditional sensory neuron *Piezo2* KO mice exhibit reduced mechanosensation and have a bladder phenotype that includes increased intercontraction interval, larger voiding pressures, and decreased urethral contraction activity [3]. Strikingly, *PIEZO2*-deficient patients also exhibit voiding defects including decreased voiding frequency, in some instances coupled with urge incontinence [3]. While a role for urothelial PIEZO2 expression cannot be excluded in these patients, one likely target is sensory neurons as *PIEZO2*-deficient individuals and mice also suffer from defects in proprioception [86–88].

Our current studies were focused on identifying a role for TMEM63B in voiding behavior. This follows up on recent studies that implicate TMEM63B in mechano- and osmo-sensation across a variety of organs, as well as our recent evidence that HA-TMEM63B is expressed within the upper and lower urinary tracts including the urothelium lining the renal pelvis, ureters, bladder, and upper urethra [48]. Our current studies further reveal that the urothelium expresses both *Tmem63a* and *Tmem63b*, and that DRG neurons, including those innervating the bladder, express HA-TMEM63B (as well as *Tmem63a* and *Tmem63b*). We further generated conditional urothelial *Tmem63b* KO mice as well as conditional sensory neuron *Tmem63b* KO mice. However, using the void spot assay, a well-documented screening tool for defects in lower urinary tract function [2,3,55–57], we did not detect a measurable phenotype resulting from loss of urothelial or sensory neuron *Tmem63b* expression. This was despite efforts to look for sex-specific differences or variations that could be ascribed to circadian effects. We also assessed whether the overactive bladder phenotype that results from treatment with CYP would be affected by loss of *Tmem63b* expression. However, we were unable to detect significant changes.

One could draw the conclusion that *Tmem63b* plays no role in normal bladder voiding behavior and responses to acute CYP treatment; however, there are several caveats worth considering. First, the screening tools and experimental conditions employed may not be sufficient to reveal a role for *Tmem63b*. Thus, by altering urine composition (e.g., making it hypertonic or hypotonic) or by exposing the bladder to bacterial infections, or chronic disease states such as partial outlet obstruction or spinal cord injury, we may reveal a role for *Tmem63b*. The second caveat is that the phenotype resulting from loss of *Tmem63b* expression in the urothelium (or sensory neurons) may be complex and involve compensatory functional changes, which may not be revealed by void spot assays. The third caveat is that there may be compensatory changes in gene expression that accompany knockout of *Tmem63b*. We noted that the urothelium and sensory neurons express both *Tmem63a* and *Tmem63b*. Thus, only a double *Tmem63a*/*Tmem63b* KO mouse may reveal a phenotype.

Such a possibility has been reported for *Tmem63a/b*-dependent mechano/osmosensation in lung epithelia [39], and as noted above, we previously reported that a bladder phenotype was only revealed in conditional urothelial *Piezo1/2* double KO mice and not individual conditional urothelial *Piezo1* or *Piezo2* KO animals [2]. An obvious next step will be to generate double *Tmem63a/b* KO animals and define their phenotype.

In sum, our current studies indicate that despite expression of TMEM63B in the urothelium and sensory neurons, loss of *Tmem63b* expression in these tissues does not result in an obvious phenotype when assessing voiding behavior. However, we cannot exclude the possibility that *Tmem63b* function will be revealed in other physiological/pathological states or when expression of both *Tmem63b* and *Tmem63a* are simultaneously knocked out.

## Acknowledgments

We thank Dr. Nicolas Montalbetti for his assistance in early pilot experiments for this project that were not included in this manuscript. The authors have nothing to report.

## Author contributions

**Conceptualization:** Marianela G. Dalghi, Yun Stone Shi, Gerard Apodaca.

**Data curation:** Marianela G. Dalghi, Dennis R. Clayton, Tanmay Parakala-Jain, Marcelo D. Carattino, Gerard Apodaca.

**Formal analysis:** Marianela G. Dalghi, Dennis R. Clayton, Tanmay Parakala-Jain, Gerard Apodaca.

**Funding acquisition:** Marcelo D. Carattino, Yun Stone Shi, Gerard Apodaca.

**Investigation:** Marianela G. Dalghi, Wily G. Ruiz, Dennis R. Clayton, Tanmay Parakala-Jain, Marcelo D. Carattino, Yun Stone Shi, Gerard Apodaca.

**Methodology:** Marianela G. Dalghi, Wily G. Ruiz, Marcelo D. Carattino, Gerard Apodaca.

**Project administration:** Gerard Apodaca.

**Resources:** Marcelo D. Carattino, Yun Stone Shi, Gerard Apodaca.

**Software:** Dennis R. Clayton, Gerard Apodaca.

**Supervision:** Gerard Apodaca.

**Validation:** Marianela G. Dalghi, Dennis R. Clayton, Gerard Apodaca.

**Visualization:** Marianela G. Dalghi, Wily G. Ruiz, Dennis R. Clayton, Marcelo D. Carattino, Gerard Apodaca.

**Writing – original draft:** Gerard Apodaca.

**Writing – review & editing:** Marianela G. Dalghi, Wily G. Ruiz, Dennis R. Clayton, Tanmay Parakala-Jain, Marcelo D. Carattino, Yun Stone Shi, Gerard Apodaca.

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
