## [Decision Letter · Decision Letter 0]

15 Jul 2025

Dear Dr. Apodaca,

We look forward to receiving your revised manuscript.

Kind regards,

Gennady S. Cymbalyuk, Ph.D.

Academic Editor

PLOS ONE

Journal Requirements:

[This work was supported by grants from the National institutes of Health including R01DK119183 (to GA and MDC), R01DK129473 (to GA), a pilot project grant supported by P30DK079307 (to MGD), and by the Pittsburgh Center for Kidney Research KIDNIT imaging core (U54DK137329). The work was also supported by grants from the National Natural Science Foundation of China 32330044 (to YSS). The Leica Stellaris confocal used in this study was funded in large part by S10OD028596 (to GA).].

[This work was supported by grants from the National institutes of Health including R01DK119183 (to GA and MDC), R01DK129473 (to GA), a pilot project grant supported by P30DK079307 (to MGD), and by the Pittsburgh Center for Kidney Research KIDNIT imaging core (U54DK137329). The work was also supported by grants from the National Natural Science Foundation of China 32330044 (to YSS). The Leica Stellaris confocal used in this study was funded in large part by S10OD028596 (to GA).]

[This work was supported by grants from the National institutes of Health including R01DK119183 (to GA and MDC), R01DK129473 (to GA), a pilot project grant supported by P30DK079307 (to MGD), and by the Pittsburgh Center for Kidney Research KIDNIT imaging core (U54DK137329). The work was also supported by grants from the National Natural Science Foundation of China 32330044 (to YSS). The Leica Stellaris confocal used in this study was funded in large part by S10OD028596 (to GA).].

Reviewers' comments:

Reviewer's Responses to Questions

**Comments to the Author**

1. Is the manuscript technically sound, and do the data support the conclusions?

Reviewer #1: No

Reviewer #2: Yes

2. Has the statistical analysis been performed appropriately and rigorously?

Reviewer #1: No

Reviewer #2: I Don't Know

3. Have the authors made all data underlying the findings in their manuscript fully available?

Reviewer #1: No

Reviewer #2: No

4. Is the manuscript presented in an intelligible fashion and written in standard English?

Reviewer #1: Yes

Reviewer #2: Yes

Reviewer #1: In this study, Dalghi et al. investigated whether TMEM63b in the urothelium and in DRG/sensory neurons regulates the voiding behavior of laboratory mice. First, the authors confirmed their published observations that in the mouse bladder, TMEM63B is dominantly expressed in the urothelium. Then, they used void spot assay to evaluate number of void spots, average void volume, and total void volume for 6 hours during the light and dark phases of conditional urothelium and conditional sensory neuron Tmem63b KO mice that were either untreated or treated with cyclophosphamide. The authors observed no differences in void spot parameters between WT and KO mice in control and CYP-treated groups.

The manuscript is well written and adds information about localization and (possible lack of) urinary function of Tmem63b in the mouse bladder urothelium and DRG. There are, however, some concerns that need to be addressed.

Major:

1. VSA data:

• The key message of this manuscript is based on observations made by a single, rather crude, method (i.e., VSA) which inherently may be unable to reveal potential differences in voiding behaviors of small sample sizes as is the case here. While VSA in mice is a popular screening tool, it is most frequently/effectively used to support arguments derived by several experimental approaches. Although their version of VSA appears to be enhanced by the video monitoring, the typical drawbacks, variables and sources of inaccuracies of the method persist.

• Variability should be reported as SD in both text and graphs (SEM is an indicator of precision, not variability).

• Considering the appearance of SEM in a number of sets in Figures 3,4,6,7, the level of reliability of the mean appears to be low. How was the sample size in the VSA studies determined?

• Quality of acclimatization is critical for obtaining reliable data. Was one-hour acclimatization sufficient to yield reproducible results in the subsequent 6 hours of testing?

• Can this VSA method discriminate between SVS or small/medium PVS and spots created by mouse footsteps spreading urine on the lining paper? Such artifacts may reflect the presence of certain urinary behaviors (e.g., due to stress, changed physical activity, pain, and others) rather than altered bladder physiology.

• How were the volumes and number of overlapping urine spots quantified?

2. RNAscope data:

• The RNAscope assay of Tmem63a and Tmem63b shown in Figure 1 suggests possible differences/gradients in the distribution of the two genes within the urothelium itself and within layers of the bladder wall. Similar ideas were alluded by the authors in reference to data shown in Figure 2. Possible distribution differences of 63a and 63b require proper analysis of the RNAscope results following the ACD guidelines and scoring system. This would provide information about the primary spatial and morphological localization of the two genes and could prevent ambiguous data interpretation. For example, on p. 23 the authors state that “the greatest signal [of Tmem63b was] in the basal cell layer.” However, the images shown in Fig. 1C appear to suggest that Tmem63b might be expressed more in intermediate and umbrella cells. If there are differences between Tmem63b and TMEM63B levels of expression, this should be supported by data analysis.

3. Animal models:

• The description of Tmem63bHA-fl/HA-fl mice and conditional urothelial Tmem63b KO mice has already been published by the authors and this should be stated explicitly. Likewise, the method of genotyping and the primers for Tmem63bHA-fl/HA-fl have been published previously. There is no need to republish the descriptions verbatim.

Minor:

- Inconsistent use of terminology: “RNAscope” in Methods, “FISH” in Results, “fluorescent in-situ hybridization” in Figure Legend. The term “RNAscope” would be most appropriate for this study.

- The number of references is excessive for a non-review article.

- p. 4: Reference 27 is not by Zhao et al.

- p.9, ln 5 from top: “hybridization” instead of hydrization

Reviewer #2: This is a straight-forward well conducted study examining the role of Tmem63b in contributing to voiding function in male and female mice. Tmem63b is a putative mechanosensitive cation channel involved in mechanosensation. In the present study, the authors used in situ hybridization to localize mRNA expression of both Tmem63a and Tmem63b to the mouse urothelium, in addition to sensory neurons in the DRG connected to bladder afferents. They took advantage of an available mouse tool that expresses a floxed Tmem63b allele with an HA tag in place of the native Tmem63b gene. They used this mouse in conjunction with immunofluorescence detection for the HA tag to show that TMEM63B protein is also expressed in the mouse urothelium and DRG. Breeding this mouse with a mouse in which Cre recombinase was driven by the Uroplakin II promoter resulted in conditional deletion of Tmem63b in the urothelium. Neither males nor females of this mouse showed any overt voiding phenotypes when examined using a void spot assay. Furthermore female mice treated with CYP to cause bladder inflammation and increased void frequency did not show any differences in voiding whether or not urothelial Tmem63b expression was present. Similar results were obtained when the Tmem63b floxed mouse was bred with a sensory nerve Cre mouse (Avil cre), in that no overt voiding phenotypes were found in males or females or after CYP challenge in the females.

The authors conclude logically that Tmem63b alone does not have a major impact on voiding function, but they concede that Tmem63a expression may compensate for the lack of Tmem63a in their mouse models. They were unable to test this hypothesis due to unavailability of appropriate Tmem63a mouse models.

Overall, this manuscript is clear and easy to follow, and the studies appear to have been conducted thoroughly and carefully. I have a few comments that the authors should address.

Major Comments:

How do you define individual voids using the void spot assay? What if multiple spots, especially secondary spots, are part of the same void? How is that handled? Since you have used video imaging to capture the data, can you use a time criteria to define voids? For example, spots occurring within a one minute time period (or some appropriate time) are considered a single void? Using video, can you then determine intermicturition intervals and report those? For phenotyping voiding behavior, both void volume and intermicturition interval should be used to get the most accurate characterization of voiding behavior. Furthermore, fluid intake should also be quantified. In the present study, the urothelial conditional knockout of Tmem63b did not appear to substantially alter voiding behavior, but it would be useful to get the most accurate depiction of voiding behavior. Any changes in voiding behavior have to be examined in the context of liquid intake, and that point is rarely considered in the literature.

Why were male mice not examined in the CYP experiment? Could there be a sex-difference in the response to CYP in urothelial Tmem63b knockout mice?

In the methods, the authors describe conducting CYP studies using both an unpaired (between subjects) and paired (within subject) design. It is not clear from the Results or Figure 4/Figure 7 which study design is being presented.

Minor Comments:

Methods, PCR analysis section, Page 8, first sentence top of page; You state that "Fine forceps were used to invert the bladder onto the pointed end of a yellow tip, trimmed by 5 mm with a scalpel, that was positioned next to the dome of the bladder." By yellow tip, do you mean a pipette tip? Please be more specific in this description about the yellow tip.

At what time were mice placed in the void spot analysis chamber for acclimation in the CYP test? Was this acclimation time kept consistent with the non-CYP treated animals?

Why was log-transformation done for analysis of the CYP dataset and not other data sets? Did distribution patterns and variances differ substantially in the different data sets? Why would this be the case?

Please define "nuc" in the figure legends.

**Do you want your identity to be public for this peer review?** For information about this choice, including consent withdrawal, please see our Privacy Policy

Reviewer #1: No

Reviewer #2: No

---

## [Author Response · Author response to Decision Letter 1]

17 Oct 2025

Responses to Reviewer #1:

Reviewer’s major comments referring to VSA data:

“The key message of this manuscript is based on observations made by a single, rather crude, method (i.e., VSA) which inherently may be unable to reveal potential differences in voiding behaviors of small sample sizes as is the case here. While VSA in mice is a popular screening tool, it is most frequently/effectively used to support arguments derived by several experimental approaches. Although their version of VSA appears to be enhanced by the video monitoring, the typical drawbacks, variables and sources of inaccuracies of the method persist.”

Response: The void spot assay (VSA), developed by Desjardins et al. (PMID: 4745598) in the 1970s, is widely adopted for its simplicity, low cost, and non-invasive nature. Importantly, in its current iteration, VSA correlates well with cystometric parameters of bladder function (see Hill et al., PMID: 30156116), particularly in studies of lower urinary tract mechanotransduction (see cited works in our manuscript) — the focus of our current work. The reviewer is correct that the traditional VSA, typically performed as an end-point assay, can suffer from limitations, including difficulties in resolving individual voiding events given the tendency of mice to urinate repeatedly in the same location. This has led investigators to water deprive their animals and shorten the time of assessment to 2-to-4 hours. Furthermore, time constraints (such as the desire to complete experiments during the workday) prevent the evaluation of mouse voiding behavior during their active night phase, lessening the ability to analyze specific genes or treatments that are governed by circadian rhythms. Additional issues are associated with discriminating small void spots (SVS) from larger ones (PVS), determining whether SVS reflect carryover of urine adhered to tails or paw, or due to post-micturition dribbling, or distinguishing if SVS are a consequence of frequent, but individual voiding events (e.g., in response to cystitis). Being able to accurately characterize SVS is relevant because conditions that result in an overactive bladder are often characterized by frequency and small void size.

The video-monitored VSA, which we employed in our studies, overcomes most of these limitations. By continuous video monitoring we can readily track the mouse behavior as the animal moves to the site of urination, pauses to release its urine, and then moves away. Importantly, we can accurately detect and measure multiple voids at the same location. Because we can provide the animals both food and water (in the form of a gel), we are able to monitor the animals’ voiding behavior over relatively long periods of time and during both the dark and light phases. Furthermore, our method allows us to readily detect SVS events, including the ability to assess whether these are associated with a defined voiding behavior (such as movement to a previously used void site, pausing to void, and then leaving), whether any trailing of urine occurs, or whether voiding is occurring as the animal is moving. These and other benefits are described in our recent JOVE publication (PMID: 36847378; reference 51 in this manuscript). We understand the Reviewer’s concern that the traditional VSA has limitations. However, our video-monitored refinements were specifically designed to address these issues, as substantiated by our recent publications (see PMID: 39684215, 37823199, 34779263, 34464353) and the current work.

Reviewer’s comment: Variability should be reported as SD in both text and graphs (SEM is an indicator of precision, not variability).

Response: Because individual data points are displayed in all of our scatter plots, the variability among animals is directly visible. Therefore, we believe the use of SEM is appropriate as it conveys the precision of our estimates of the mean. If we understand correctly, PLOS ONE accepts either SD or SEM when reporting data.

Reviewer’s comment: Considering the appearance of SEM in a number of sets in Figures 3,4,6,7, the level of reliability of the mean appears to be low. How was the sample size in the VSA studies determined?

Response: Our sample sizes were based on our previous experience performing these assays and multiple other reports in the literature. We have added an additional number of animals in Fig. 3A. Please note that since the initial review, we were only able to generate a relatively small number of male Tmem63b;Avil-Cre mice and that limited our cyclophosphamide experiments to n=4.

Reviewer’s comment: Quality of acclimatization is critical for obtaining reliable data. Was one-hour acclimatization sufficient to yield reproducible results in the subsequent 6 hours of testing?

Response: Data on the role of acclimatization in VSA is mixed with some studies reporting no effect of acclimatization (even after 5 days; PMID 24717733), while others showing an effect. Many studies include no acclimatization period in their experimental design. In our experiments, we provide a defined amount of acclimatization for all animals and this was replicated across genotype and treatment group. This method has been used in several of our recent studies (see PMID: 39684215, 37823199, 34779263, 34464353), and we have been able to detect significant changes in voiding behavior under several different conditions.

Reviewer’s comment: Can this VSA method discriminate between SVS or small/medium PVS and spots created by mouse footsteps spreading urine on the lining paper? Such artifacts may reflect the presence of certain urinary behaviors (e.g., due to stress, changed physical activity, pain, and others) rather than altered bladder physiology.

Response: Using a standard curve of spotted urine, we can reliably detect voiding events as small as 2 microliters. As noted above, and because all events are monitored continuously, we can assess whether SVS and PVS are associated with a defined voiding behavior (such as movement to a void site, previously used or not, pausing to void, and then leaving). Continuous video monitoring allows us to determine whether SVS or PVS correspond to defined voiding behaviors, assess trailing of urine, and identify voiding during movement. To ensure conservative interpretation, we classify all spotting events occurring within 60 seconds as a single voiding event.

Reviewer’s comment: How were the volumes and number of overlapping urine spots quantified?

Response: As noted above, and mentioned in our Methods, volumes are derived from standard curves of spotted urine. In our system, the void spots fluoresce brightly during voiding but then fade rapidly over the next few minutes. Thus, we can readily detect multiple voids even when they occur in the same location. All of these methods are described with enormous detail in our previously published JOVE manuscript and associated video (PMID: 36847378; reference 51 in this manuscript).

Reviewer’s major comments referring to RNAscope data:

The RNAscope assay of Tmem63a and Tmem63b shown in Figure 1 suggests possible differences/gradients in the distribution of the two genes within the urothelium itself and within layers of the bladder wall. Similar ideas were alluded by the authors in reference to data shown in Figure 2. Possible distribution differences of 63a and 63b require proper analysis of the RNAscope results following the ACD guidelines and scoring system. This would provide information about the primary spatial and morphological localization of the two genes and could prevent ambiguous data interpretation. For example, on p. 23 the authors state that “the greatest signal [of Tmem63b was] in the basal cell layer.” However, the images shown in Fig. 1C appear to suggest that Tmem63b might be expressed more in intermediate and umbrella cells. If there are differences between Tmem63b and TMEM63B levels of expression, this should be supported by data analysis.

Response: The ACD guidelines and scoring system is a semiquantitative approach that depends on binning of signal dot data. We have taken a more quantitative approach, detailed in the Methods section, whereby signal dots are counted in random urothelial cells (nuclei) including those in the basal, intermediate, and umbrella cell layers. We also assessed whether urothelial cells expressed either or both Tmem63a and Tmem63b. Our new experiments confirm that all urothelial cells co-express Tmem63a and Tmem63b. These new data are incorporated into revised Fig. 1D and fit well with the immunofluorescence images presented in Fig. 1C.

Given the well-documented disconnect between transcription and translation, and the possibility of differential protein turnover, it is not completely surprising that TMEM63B protein expression is not identical to expression of Tmem63b message. Indeed, we observe that TMEM63B protein expression is visually highest in the basal cell layer and decreases as one progresses to the umbrella cell layer (see Fig. 2).

Reviewer’s major comments referring to animal models:

The description of Tmem63bHA-fl/HA-fl mice and conditional urothelial Tmem63b KO mice has already been published by the authors and this should be stated explicitly. Likewise, the method of genotyping and the primers for Tmem63bHA-fl/HA-fl have been published previously. There is no need to republish the descriptions verbatim.

Response: We have edited the revised manuscript to exclude a description of the genotyping and primers as requested. We also make sure to cite our previous manuscript describing some of these mice. Although, it is worth noting that the previous characterization of the conditional KO mice was minimal and did not include an assessment of knockout efficiency.

Reviewer’s minor comments:

- Inconsistent use of terminology: “RNAscope” in Methods, “FISH” in Results, “fluorescent in-situ hybridization” in Figure Legend. The term “RNAscope” would be most appropriate for this study.

Response: As suggested, we now use the term RNAscope throughout.

- The number of references is excessive for a non-review article.

Response: We appreciate the Reviewer’s observation regarding the number of references. However, we believe that each cited work provides essential methodological or conceptual context, and thus contributes directly to the manuscript’s rigor.

- p. 4: Reference 27 is not by Zhao et al.

Response: The offending reference is removed from this section.

- p.9, ln 5 from top: “hybridization” instead of hydrization

Response: Thank you for catching this – the typo has been corrected.

Responses to Reviewer #2:

Reviewer’s major comments about the void spot assay:

How do you define individual voids using the void spot assay? What if multiple spots, especially secondary spots, are part of the same void? How is that handled? Since you have used video imaging to capture the data, can you use a time criteria to define voids? For example, spots occurring within a one minute time period (or some appropriate time) are considered a single void? Using video, can you then determine intermicturition intervals and report those? For phenotyping voiding behavior, both void volume and intermicturition interval should be used to get the most accurate characterization of voiding behavior. Furthermore, fluid intake should also be quantified. In the present study, the urothelial conditional knockout of Tmem63b did not appear to substantially alter voiding behavior, but it would be useful to get the most accurate depiction of voiding behavior. Any changes in voiding behavior have to be examined in the context of liquid intake, and that point is rarely considered in the literature.

Response: Please see our responses to reviewer #1 concerning identifying primary and secondary void spots, and use of a one-minute rule when characterizing these events. These are all detailed in our previous manuscripts including a detailed JOVE protocol (PMID: 36847378; reference 51 in this manuscript). The reviewer suggested that we consider including a description of intermicturition interval, a parameter usually associated with cystometry. After some thought, we decided that measuring frequency (number of total voids/h) would provide a similar type of characterization. This data is now included for each of the VSA figures and substitutes for our previous measure of continence. In the case of female mice, we observed the expected increase in frequency associated with cyclophosphamide-induced cystitis, a validation of this approach. The reviewer also asked us to measure fluid intake, something rarely done in the VSA because mice often disturb/play with their water source. In our case, the relatively long periods of analysis required that we include some form of water, which we provided in gel form. Unfortunately, and despite several attempts to quantify water intake, it was not possible. Perhaps reflecting the weight of the water gel (about 60g; animals weigh 20-25 g), variations in humidity, animal sex, and circadian effects, we could not detect reproducible changes when we compared the weight of water gel in the presence or absence of the animals. However, we should note that by adding up the total PVS and total SVS volume once can estimate the animal’s water intake.

Reviewer’s major comments about CYP experiments in male mice:

Why were male mice not examined in the CYP experiment? Could there be a sex-difference in the response to CYP in urothelial Tmem63b knockout mice?

Response: At the reviewer’s request, we have added male CYP experiments for both Tmem63b;Upk2-Cre and Tmem63b;Avil-Cre mice. Interestingly, at 150 mg/kg, cyclophosphamide did not generate a significant effect. This may indicate that male mice are resistant to this dose of cyclophosphamide. Please note that we were only able to generate a relatively small number of male Tmem63b;Avil-Cre mice and this limited our cyclophosphamide experiments to n=4 for this genotype.

Reviewer’s major comments about paired vs unpaired experiments:

In the methods, the authors describe conducting CYP studies using both an unpaired (between subjects) and paired (within subject) design. It is not clear from the Results or Figure 4/Figure 7 which study design is being presented.

Response: In the revised manuscript we have made it more clear in the figure legends for Fig. 4 and Fig. 7 which experiments were paired and which were unpaired.

Reviewer’s Minor Comments:

Methods, PCR analysis section, Page 8, first sentence top of page; You state that "Fine forceps were used to invert the bladder onto the pointed end of a yellow tip, trimmed by 5 mm with a scalpel, that was positioned next to the dome of the bladder." By yellow tip, do you mean a pipette tip? Please be more specific in this description about the yellow tip.

Response: Yes, it was a 200-microliter yellow pipette tip. We have included this information in the revised manuscript.

At what time were mice placed in the void spot analysis chamber for acclimation in the CYP test? Was this acclimation time kept consistent with the non-CYP treated animals?

Response: This is described in the methods section and was identical for mice irrespective of genotype or sex.

Why was log-transformation done for analysis of the CYP dataset and not other data sets? Did distribution patterns and variances differ substantially in the different data sets? Why would this be the case?

Response: The description of log transformation was an unfortunate carryover from a draft version of the manuscript. In the revised manuscript, all data was untransformed and analyzed identically.

Please define "nuc" in the figure legends.

Response: Nuclei (nuc) is now defined in the relevant legends for figures 1, 2, and 5.

---

## [Decision Letter · Decision Letter 1]

2 Nov 2025

Conditional deletion of Tmem63b does not impact mouse voiding behavior

PONE-D-25-20430R1

Dear Dr. Apodaca,

We’re pleased to inform you that your manuscript has been judged scientifically suitable for publication and will be formally accepted for publication once it meets all outstanding technical requirements.

Kind regards,

Gennady S. Cymbalyuk, Ph.D.

Academic Editor

PLOS ONE

Additional Editor Comments (optional):

Reviewers' comments:

Reviewer's Responses to Questions

**Comments to the Author**

Reviewer #1: All comments have been addressed

Reviewer #2: All comments have been addressed

2. Is the manuscript technically sound, and do the data support the conclusions?

Reviewer #1: Partly

Reviewer #2: (No Response)

3. Has the statistical analysis been performed appropriately and rigorously?

Reviewer #1: Yes

Reviewer #2: (No Response)

4. Have the authors made all data underlying the findings in their manuscript fully available?

Reviewer #1: No

Reviewer #2: (No Response)

5. Is the manuscript presented in an intelligible fashion and written in standard English?

Reviewer #1: Yes

Reviewer #2: (No Response)

Reviewer #1: (No Response)

Reviewer #2: (No Response)

**Do you want your identity to be public for this peer review?** For information about this choice, including consent withdrawal, please see our Privacy Policy

Reviewer #1: No

Reviewer #2: No

---

## [Editor Report · Acceptance letter]

PONE-D-25-20430R1

PLOS ONE

Dear Dr. Apodaca,

I'm pleased to inform you that your manuscript has been deemed suitable for publication in PLOS ONE. Congratulations! Your manuscript is now being handed over to our production team.

Kind regards,

on behalf of

Dr. Gennady S. Cymbalyuk

Academic Editor

PLOS ONE